# Measuring Sample Quality with Stein's Method

**Jackson Gorham**
Department of Statistics
Stanford University

**Lester Mackey**
Department of Statistics
Stanford University

## Abstract

To improve the efficiency of Monte Carlo estimation, practitioners are turning to biased Markov chain Monte Carlo procedures that trade off asymptotic exactness for computational speed. The reasoning is sound: a reduction in variance due to more rapid sampling can outweigh the bias introduced. However, the inexactness creates new challenges for sampler and parameter selection, since standard measures of sample quality like effective sample size do not account for asymptotic bias. To address these challenges, we introduce a new computable quality measure based on Stein's method that bounds the discrepancy between sample and target expectations over a large class of test functions. We use our tool to compare exact, biased, and deterministic sample sequences and illustrate applications to hyper-parameter selection, convergence rate assessment, and quantifying bias-variance tradeoffs in posterior inference.

## 1 Introduction

When faced with a complex target distribution, one often turns to Markov chain Monte Carlo (MCMC) [1] to approximate intractable expectations $\mathbb{E}_P[h(Z)] = \int_{\mathcal{X}} p(x)h(x)dx$ with asymptotically exact sample estimates $\mathbb{E}_Q[h(X)] = \sum_{i=1}^n q(x_i)h(x_i)$. These complex targets commonly arise as posterior distributions in Bayesian inference and as candidate distributions in maximum likelihood estimation [2]. In recent years, researchers [e.g., 3, 4, 5] have introduced asymptotic bias into MCMC procedures to trade off asymptotic correctness for improved sampling speed. The rationale is that more rapid sampling can reduce the variance of a Monte Carlo estimate and hence outweigh the bias introduced. However, the added flexibility introduces new challenges for sampler and parameter selection, since standard sample quality measures, like effective sample size, asymptotic variance, trace and mean plots, and pooled and within-chain variance diagnostics, presume eventual convergence to the target [1] and hence do not account for asymptotic bias.

To address this shortcoming, we develop a new measure of sample quality suitable for comparing asymptotically exact, asymptotically biased, and even deterministic sample sequences. The quality measure is based on Stein's method and is attainable by solving a linear program. After outlining our design criteria in Section 2, we relate the convergence of the quality measure to that of standard probability metrics in Section 3, develop a streamlined implementation based on geometric spanners in Section 4, and illustrate applications to hyperparameter selection, convergence rate assessment, and the quantification of bias-variance tradeoffs in posterior inference in Section 5. We discuss related work in Section 6 and defer all proofs to the appendix.

**Notation** We denote the $\ell_2$, $\ell_1$, and $\ell_\infty$ norms on $\mathbb{R}^d$ by $\|\cdot\|_2$, $\|\cdot\|_1$, and $\|\cdot\|_\infty$ respectively. We will often refer to a generic norm $\|\cdot\|$ on $\mathbb{R}^d$ with associated dual norms $\|w\|^* \triangleq \sup_{v \in \mathbb{R}^d : \|v\|=1} \langle w, v \rangle$ for vectors $w \in \mathbb{R}^d$, $\|M\|^* \triangleq \sup_{v \in \mathbb{R}^d : \|v\|=1} \|Mv\|^*$ for matrices $M \in \mathbb{R}^{d \times d}$, and $\|T\|^* \triangleq \sup_{v \in \mathbb{R}^d : \|v\|=1} \|T[v]\|^*$ for tensors $T \in \mathbb{R}^{d \times d \times d}$. We denote the $j$-th standard basis vector by $e_j$, the partial derivative $\frac{\partial}{\partial x_k}$ by $\nabla_k$, and the gradient of any $\mathbb{R}^d$-valued function $g$ by $\nabla g$ with components $(\nabla g(x))_{jk} \triangleq \nabla_k g_j(x)$.

## 2 Quality Measures for Samples

Consider a target distribution $P$ with open convex support $\mathcal{X} \subseteq \mathbb{R}^d$ and continuously differentiable density $p$. We assume that $p$ is known up to its normalizing constant and that exact integration under $P$ is intractable for most functions of interest. We will approximate expectations under $P$ with the aid of a *weighted sample*, a collection of distinct sample points $x_1, \ldots, x_n \in \mathcal{X}$ with weights $q(x_i)$ encoded in a probability mass function $q$. The probability mass function $q$ induces a discrete distribution $Q$ and an approximation $\mathbb{E}_Q[h(X)] = \sum_{i=1}^n q(x_i)h(x_i)$ for any target expectation $\mathbb{E}_P[h(Z)]$. We make no assumption about the provenance of the sample points; they may arise as random draws from a Markov chain or even be deterministically selected.

Our goal is to compare the fidelity of different samples approximating a common target distribution. That is, we seek to quantify the discrepancy between $\mathbb{E}_Q$ and $\mathbb{E}_P$ in a manner that (i) detects when a sequence of samples is converging to the target, (ii) detects when a sequence of samples is not converging to the target, and (iii) is computationally feasible. A natural starting point is to consider the maximum deviation between sample and target expectations over a class of real-valued test functions $\mathcal{H}$,

$$d_{\mathcal{H}}(Q, P) = \sup_{h \in \mathcal{H}} |\mathbb{E}_Q[h(X)] - \mathbb{E}_P[h(Z)]|. \tag{1}$$

When the class of test functions is sufficiently large, the convergence of $d_{\mathcal{H}}(Q_m, P)$ to zero implies that the sequence of sample measures $(Q_m)_{m \geq 1}$ converges weakly to $P$. In this case, the expression (1) is termed an *integral probability metric* (IPM) [6]. By varying the class of test functions $\mathcal{H}$, we can recover many well-known probability metrics as IPMs, including the *total variation distance*, generated by $\mathcal{H} = \{h : \mathcal{X} \to \mathbb{R} \mid \sup_{x \in \mathcal{X}} |h(x)| \leq 1\}$, and the *Wasserstein distance* (also known as the Kantorovich-Rubenstein or earth mover's distance), $d_{\mathcal{W}_{\|\cdot\|}}$, generated by

$$\mathcal{H} = \mathcal{W}_{\|\cdot\|} \triangleq \{h : \mathcal{X} \to \mathbb{R} \mid \sup_{x \neq y \in \mathcal{X}} \tfrac{|h(x) - h(y)|}{\|x - y\|} \leq 1\}.$$

The primary impediment to adopting an IPM as a sample quality measure is that exact computation is typically infeasible when generic integration under $P$ is intractable. However, we could skirt this intractability by focusing on classes of test functions with known expectation under $P$. For example, if we consider only test functions $h$ for which $\mathbb{E}_P[h(Z)] = 0$, then the IPM value $d_{\mathcal{H}}(Q, P)$ is the solution of an optimization problem depending on $Q$ alone. This, at a high level, is our strategy, but many questions remain. How do we select the class of test functions $h$? How do we know that the resulting IPM will track convergence and non-convergence of a sample sequence (Desiderata (i) and (ii))? How do we solve the resulting optimization problem in practice (Desideratum (iii))? To address the first two of these questions, we draw upon tools from Charles Stein's method of characterizing distributional convergence. We return to the third question in Section 4.

## 3 Stein's Method

Stein's method [7] for characterizing convergence in distribution classically proceeds in three steps:

1. Identify a real-valued operator $\mathcal{T}$ acting on a set $\mathcal{G}$ of $\mathbb{R}^d$-valued[1] functions of $\mathcal{X}$ for which

$$\mathbb{E}_P[(\mathcal{T}g)(Z)] = 0 \quad \text{for all} \quad g \in \mathcal{G}. \tag{2}$$

   Together, $\mathcal{T}$ and $\mathcal{G}$ define the *Stein discrepancy*,

$$\mathcal{S}(Q, \mathcal{T}, \mathcal{G}) \triangleq \sup_{g \in \mathcal{G}} |\mathbb{E}_Q[(\mathcal{T}g)(X)]| = \sup_{g \in \mathcal{G}} |\mathbb{E}_Q[(\mathcal{T}g)(X)] - \mathbb{E}_P[(\mathcal{T}g)(Z)]| = d_{\mathcal{T}\mathcal{G}}(Q, P),$$

   an IPM-type quality measure with no explicit integration under $P$.

2. Lower bound the Stein discrepancy by a familiar convergence-determining IPM $d_{\mathcal{H}}$. This step can be performed once, in advance, for large classes of target distributions and ensures that, for any sequence of probability measures $(\mu_m)_{m \geq 1}$, $\mathcal{S}(\mu_m, \mathcal{T}, \mathcal{G})$ converges to zero only if $d_{\mathcal{H}}(\mu_m, P)$ does (Desideratum (ii)).

3. Upper bound the Stein discrepancy by any means necessary to demonstrate convergence to zero under suitable conditions (Desideratum (i)). In our case, the universal bound established in Section 3.3 will suffice.

While Stein's method is typically employed as an analytical tool, we view the Stein discrepancy as a promising candidate for a practical sample quality measure. Indeed, in Section 4, we will adopt an optimization perspective and develop efficient procedures to compute the Stein discrepancy for any sample measure $Q$ and appropriate choices of $\mathcal{T}$ and $\mathcal{G}$. First, we assess the convergence properties of an equivalent Stein discrepancy in the subsections to follow.

## 3.1 Identifying a Stein Operator

The *generator method* of Barbour [8] provides a convenient and general means of constructing operators $\mathcal{T}$ which produce mean-zero functions under $P$ (2) . Let $(Z_t)_{t \geq 0}$ represent a Markov process with unique stationary distribution $P$. Then the *infinitesimal generator* $\mathcal{A}$ of $(Z_t)_{t \geq 0}$, defined by

$$(\mathcal{A}u)(x) = \lim_{t \to 0} \left( \mathbb{E}[u(Z_t) \mid Z_0 = x] - u(x) \right)/t \quad \text{for} \quad u : \mathbb{R}^d \to \mathbb{R},$$

satisfies $\mathbb{E}_P[(\mathcal{A}u)(Z)] = 0$ under mild conditions on $\mathcal{A}$ and $u$. Hence, a candidate operator $\mathcal{T}$ can be constructed from any infinitesimal generator.

For example, the *overdamped Langevin diffusion*, defined by the stochastic differential equation $dZ_t = \frac{1}{2} \nabla \log p(Z_t) dt + dW_t$ for $(W_t)_{t \geq 0}$ a Wiener process, gives rise to the generator

$$(\mathcal{A}_P u)(x) = \frac{1}{2} \langle \nabla u(x), \nabla \log p(x) \rangle + \frac{1}{2} \langle \nabla, \nabla u(x) \rangle. \tag{3}$$

After substituting $g$ for $\frac{1}{2} \nabla u$, we obtain the associated *Stein operator*

$$(\mathcal{T}_P g)(x) \triangleq \langle g(x), \nabla \log p(x) \rangle + \langle \nabla, g(x) \rangle. \tag{4}$$

The Stein operator $\mathcal{T}_P$ is particularly well-suited to our setting as it depends on $P$ only through the derivative of its log density and hence is computable even when the normalizing constant of $p$ is not.

If we let $\partial \mathcal{X}$ denote the boundary of $\mathcal{X}$ (an empty set when $\mathcal{X} = \mathbb{R}^d$) and $n(x)$ represent the outward unit normal vector to the boundary at $x$, then we may define the *classical Stein set*

$$\mathcal{G}_{\|\cdot\|} \triangleq \left\{ g : \mathcal{X} \to \mathbb{R}^d \; \middle| \; \sup_{x \neq y \in \mathcal{X}} \max \left( \|g(x)\|^*, \|\nabla g(x)\|^*, \frac{\|\nabla g(x) - \nabla g(y)\|^*}{\|x - y\|} \right) \leq 1 \quad \text{and} \right.$$

$$\left. \langle g(x), n(x) \rangle = 0, \forall x \in \partial \mathcal{X} \text{ with } n(x) \text{ defined} \right\}$$

of sufficiently smooth functions satisfying a Neumann-type boundary condition. The following proposition – a consequence of integration by parts – shows that $\mathcal{G}_{\|\cdot\|}$ is a suitable domain for $\mathcal{T}_P$.

**Proposition 1.** *If* $\mathbb{E}_P[\|\nabla \log p(Z)\|] < \infty$, *then* $\mathbb{E}_P[(\mathcal{T}_P g)(Z)] = 0$ *for all* $g \in \mathcal{G}_{\|\cdot\|}$.

Together, $\mathcal{T}_P$ and $\mathcal{G}_{\|\cdot\|}$ form the *classical Stein discrepancy* $\mathcal{S}(Q, \mathcal{T}_P, \mathcal{G}_{\|\cdot\|})$, our chief object of study.

## 3.2 Lower Bounding the Classical Stein Discrepancy

In the univariate setting ($d = 1$), it is known for a wide variety of targets $P$ that the classical Stein discrepancy $\mathcal{S}(\mu_m, \mathcal{T}_P, \mathcal{G}_{\|\cdot\|})$ converges to zero only if the Wasserstein distance $d_{\mathcal{W}_{\|\cdot\|}}(\mu_m, P)$ does [9, 10]. In the multivariate setting, analogous statements are available for multivariate Gaussian targets [11, 12, 13], but few other target distributions have been analyzed. To extend the reach of the multivariate literature, we show in Theorem 2 that the classical Stein discrepancy also determines Wasserstein convergence for a large class of strongly log-concave densities, including the Bayesian logistic regression posterior under Gaussian priors.

**Theorem 2** (Stein Discrepancy Lower Bound for Strongly Log-concave Densities)**.** *If* $\mathcal{X} = \mathbb{R}^d$, *and* $\log p$ *is strongly concave with third and fourth derivatives bounded and continuous, then, for any probability measures* $(\mu_m)_{m \geq 1}$, $\mathcal{S}(\mu_m, \mathcal{T}_P, \mathcal{G}_{\|\cdot\|}) \to 0$ *only if* $d_{\mathcal{W}_{\|\cdot\|}}(\mu_m, P) \to 0$.

We emphasize that the sufficient conditions in Theorem 2 are certainly not necessary for lower bounding the classical Stein discrepancy. We hope that the theorem and its proof will provide a template for lower bounding $\mathcal{S}(Q, \mathcal{T}_P, \mathcal{G}_{\|\cdot\|})$ for other large classes of multivariate target distributions.

### 3.3 Upper Bounding the Classical Stein Discrepancy

We next establish sufficient conditions for the convergence of the classical Stein discrepancy to zero.

**Proposition 3** (Stein Discrepancy Upper Bound). *If $X \sim Q$ and $Z \sim P$ with $\nabla \log p(Z)$ integrable,*

$$\mathcal{S}(Q, \mathcal{T}_P, \mathcal{G}_{\|\cdot\|}) \leq \mathbb{E}[\|X - Z\|] + \mathbb{E}[\|\nabla \log p(X) - \nabla \log p(Z)\|] + \mathbb{E}[\|\nabla \log p(Z)(X - Z)^\top\|]$$

$$\leq \mathbb{E}[\|X - Z\|] + \mathbb{E}[\|\nabla \log p(X) - \nabla \log p(Z)\|] + \sqrt{\mathbb{E}[\|\nabla \log p(Z)\|^2] \mathbb{E}[\|X - Z\|^2]}.$$

One implication of Proposition 3 is that $\mathcal{S}(Q_m, \mathcal{T}_P, \mathcal{G}_{\|\cdot\|})$ converges to zero whenever $X_m \sim Q_m$ converges in mean-square to $Z \sim P$ and $\nabla \log p(X_m)$ converges in mean to $\nabla \log p(Z)$.

### 3.4 Extension to Non-uniform Stein Sets

The analyses and algorithms in this paper readily accommodate non-uniform Stein sets of the form

$$\mathcal{G}_{\|\cdot\|}^{c_{1:3}} \triangleq \left\{ g : \mathcal{X} \to \mathbb{R}^d \;\middle|\; \begin{array}{l} \sup_{x \neq y \in \mathcal{X}} \max\left( \frac{\|g(x)\|^*}{c_1}, \frac{\|\nabla g(x)\|^*}{c_2}, \frac{\|\nabla g(x) - \nabla g(y)\|^*}{c_3 \|x - y\|} \right) \leq 1 \text{ and} \\ \langle g(x), n(x) \rangle = 0, \forall x \in \partial \mathcal{X} \text{ with } n(x) \text{ defined} \end{array} \right\} \quad (5)$$

for constants $c_1, c_2, c_3 > 0$ known as *Stein factors* in the literature. We will exploit this additional flexibility in Section 5.2 to establish tight lower-bounding relations between the Stein discrepancy and Wasserstein distance for well-studied target distributions. For general use, however, we advocate the parameter-free classical Stein set and graph Stein sets to be introduced in the sequel. Indeed, any non-uniform Stein discrepancy is equivalent to the classical Stein discrepancy in a strong sense:

**Proposition 4** (Equivalence of Non-uniform Stein Discrepancies). *For any $c_1, c_2, c_3 > 0$,*

$$\min(c_1, c_2, c_3) \mathcal{S}(Q, \mathcal{T}_P, \mathcal{G}_{\|\cdot\|}) \leq \mathcal{S}(Q, \mathcal{T}_P, \mathcal{G}_{\|\cdot\|}^{c_{1:3}}) \leq \max(c_1, c_2, c_3) \mathcal{S}(Q, \mathcal{T}_P, \mathcal{G}_{\|\cdot\|}).$$

## 4 Computing Stein Discrepancies

In this section, we introduce an efficiently computable Stein discrepancy with convergence properties equivalent to those of the classical discrepancy. We restrict attention to the unconstrained domain $\mathcal{X} = \mathbb{R}^d$ in Sections 4.1-4.3 and present extensions for constrained domains in Section 4.4.

### 4.1 Graph Stein Discrepancies

Evaluating a Stein discrepancy $\mathcal{S}(Q, \mathcal{T}_P, \mathcal{G})$ for a fixed $(Q, P)$ pair reduces to solving an optimization program over functions $g \in \mathcal{G}$. For example, the classical Stein discrepancy is the optimum

$$\mathcal{S}(Q, \mathcal{T}_P, \mathcal{G}_{\|\cdot\|}) = \sup_g \sum_{i=1}^n q(x_i)(\langle g(x_i), \nabla \log p(x_i) \rangle + \langle \nabla, g(x_i) \rangle) \quad (6)$$

$$\text{s.t. } \|g(x)\|^* \leq 1, \|\nabla g(x)\|^* \leq 1, \|\nabla g(x) - \nabla g(y)\|^* \leq \|x - y\|, \forall x, y \in \mathcal{X}.$$

Note that the objective associated with any Stein discrepancy $\mathcal{S}(Q, \mathcal{T}_P, \mathcal{G})$ is linear in $g$ and, since $Q$ is discrete, only depends on $g$ and $\nabla g$ through their values at each of the $n$ sample points $x_i$. The primary difficulty in solving the classical Stein program (6) stems from the infinitude of constraints imposed by the classical Stein set $\mathcal{G}_{\|\cdot\|}$. One way to avoid this difficulty is to impose the classical smoothness constraints at only a finite collection of points. To this end, for each finite graph $G = (V, E)$ with vertices $V \subset \mathcal{X}$ and edges $E \subset V^2$, we define the *graph Stein set*,

$$\mathcal{G}_{\|\cdot\|, Q, G} \triangleq \left\{ g : \mathcal{X} \to \mathbb{R}^d \;|\; \forall x \in V, \; \max(\|g(x)\|^*, \|\nabla g(x)\|^*) \leq 1 \text{ and}, \forall (x, y) \in E, \right.$$

$$\left. \max\left( \frac{\|g(x) - g(y)\|^*}{\|x - y\|}, \frac{\|\nabla g(x) - \nabla g(y)\|^*}{\|x - y\|}, \frac{\|g(x) - g(y) - \nabla g(x)(x - y)\|^*}{\frac{1}{2}\|x - y\|^2}, \frac{\|g(x) - g(y) - \nabla g(y)(x - y)\|^*}{\frac{1}{2}\|x - y\|^2} \right) \leq 1 \right\},$$

the family of functions which satisfy the classical constraints and certain implied Taylor compatibility constraints at pairs of points in $E$. Remarkably, if the graph $G_1$ consists of edges between all distinct sample points $x_i$, then the associated *complete graph Stein discrepancy* $\mathcal{S}(Q, \mathcal{T}_P, \mathcal{G}_{\|\cdot\|, Q, G_1})$ is equivalent to the classical Stein discrepancy in the following strong sense.

**Proposition 5** (Equivalence of Classical and Complete Graph Stein Discrepancies). *If $\mathcal{X} = \mathbb{R}^d$, and $G_1 = (\mathrm{supp}(Q), E_1)$ with $E_1 = \{(x_i, x_l) \in \mathrm{supp}(Q)^2 : x_i \neq x_l\}$, then*
$$\mathcal{S}(Q, \mathcal{T}_P, \mathcal{G}_{\|\cdot\|}) \leq \mathcal{S}(Q, \mathcal{T}_P, \mathcal{G}_{\|\cdot\|, Q, G_1}) \leq \kappa_d \, \mathcal{S}(Q, \mathcal{T}_P, \mathcal{G}_{\|\cdot\|}),$$
*where $\kappa_d$ is a constant, independent of $(Q, P)$, depending only on the dimension $d$ and norm $\|\cdot\|$.*

Proposition 5 follows from the Whitney-Glaeser extension theorem for smooth functions [14, 15] and implies that the complete graph Stein discrepancy inherits all of the desirable convergence properties of the classical discrepancy. However, the complete graph also introduces order $n^2$ constraints, rendering computation infeasible for large samples. To achieve the same form of equivalence while enforcing only $O(n)$ constraints, we will make use of sparse *geometric spanner* subgraphs.

## 4.2 Geometric Spanners

For a given dilation factor $t \geq 1$, a *t-spanner* [16, 17] is a graph $G = (V, E)$ with weight $\|x - y\|$ on each edge $(x, y) \in E$ and a path between each pair $x' \neq y' \in V$ with total weight no larger than $t\|x' - y'\|$. The next proposition shows that *spanner Stein discrepancies* enjoy the same convergence properties as the complete graph Stein discrepancy.

**Proposition 6** (Equivalence of Spanner and Complete Graph Stein Discrepancies). *If $\mathcal{X} = \mathbb{R}^d$, $G_t = (\mathrm{supp}(Q), E)$ is a t-spanner, and $G_1 = (\mathrm{supp}(Q), \{(x_i, x_l) \in \mathrm{supp}(Q)^2 : x_i \neq x_l\})$, then*
$$\mathcal{S}(Q, \mathcal{T}_P, \mathcal{G}_{\|\cdot\|, Q, G_1}) \leq \mathcal{S}(Q, \mathcal{T}_P, \mathcal{G}_{\|\cdot\|, Q, G_t}) \leq 2t^2 \, \mathcal{S}(Q, \mathcal{T}_P, \mathcal{G}_{\|\cdot\|, Q, G_1}).$$

Moreover, for any $\ell_p$ norm, a 2-spanner with $O(\kappa_d n)$ edges can be computed in $O(\kappa_d n \log(n))$ expected time for $\kappa_d$ a constant depending only on $d$ and $\|\cdot\|$ [18]. As a result, we will adopt a 2-spanner Stein discrepancy, $\mathcal{S}(Q, \mathcal{T}_P, \mathcal{G}_{\|\cdot\|, Q, G_2})$, as our standard quality measure.

## 4.3 Decoupled Linear Programs

The final unspecified component of our Stein discrepancy is the choice of norm $\|\cdot\|$. We recommend the $\ell_1$ norm, as the resulting optimization problem decouples into $d$ independent finite-dimensional linear programs (LPs) that can be solved in parallel. More precisely, $\mathcal{S}(Q, \mathcal{T}_P, \mathcal{G}_{\|\cdot\|_1, Q, (V,E)})$ equals

$$\sum_{j=1}^{d} \sup_{\gamma_j \in \mathbb{R}^{|V|}, \Gamma_j \in \mathbb{R}^{d \times |V|}} \sum_{i=1}^{|V|} q(v_i)(\gamma_{ji} \nabla_j \log p(v_i) + \Gamma_{jji}) \tag{7}$$

$$\text{s.t. } \|\gamma_j\|_\infty \leq 1, \|\Gamma_j\|_\infty \leq 1, \text{ and } \forall\, i \neq l : (v_i, v_l) \in E,$$

$$\max\left( \frac{|\gamma_{ji} - \gamma_{jl}|}{\|v_i - v_l\|_1}, \frac{\|\Gamma_j(e_i - e_l)\|_\infty}{\|v_i - v_l\|_1}, \frac{|\gamma_{ji} - \gamma_{jl} - \langle \Gamma_j e_i, v_i - v_l \rangle|}{\frac{1}{2}\|v_i - v_l\|_1^2}, \frac{|\gamma_{ji} - \gamma_{jl} - \langle \Gamma_j e_l, v_i - v_l \rangle|}{\frac{1}{2}\|v_i - v_l\|_1^2} \right) \leq 1.$$

We have arbitrarily numbered the elements $v_i$ of the vertex set $V$ so that $\gamma_{ji}$ represents the function value $g_j(v_i)$, and $\Gamma_{jki}$ represents the gradient value $\nabla_k g_j(v_i)$.

## 4.4 Constrained Domains

A small modification to the unconstrained formulation (7) extends our tractable Stein discrepancy computation to any domain defined by coordinate boundary constraints, that is, to $\mathcal{X} = (\alpha_1, \beta_1) \times \cdots \times (\alpha_d, \beta_d)$ with $-\infty \leq \alpha_j < \beta_j \leq \infty$ for all $j$. Specifically, for each dimension $j$, we augment the $j$-th coordinate linear program of (7) with the boundary compatibility constraints

$$\max\left( \frac{|\gamma_{ji}|}{|v_{ij} - b_j|}, \frac{|\Gamma_{jki}|}{|v_{ij} - b_j|}, \frac{|\gamma_{ji} - \Gamma_{jji}(v_{ij} - b_j)|}{\frac{1}{2}(v_{ij} - b_j)^2} \right) \leq 1, \text{ for each } i,\ b_j \in \{\alpha_j, \beta_j\} \cap \mathbb{R}, \text{ and } k \neq j. \tag{8}$$

These additional constraints ensure that our candidate function and gradient values can be extended to a smooth function satisfying the boundary conditions $\langle g(z), n(z) \rangle = 0$ on $\partial\mathcal{X}$. Proposition 15 in the appendix shows that the spanner Stein discrepancy so computed is strongly equivalent to the classical Stein discrepancy on $\mathcal{X}$.

Algorithm 1 summarizes the complete solution for computing our recommended, parameter-free spanner Stein discrepancy in the multivariate setting. Notably, the spanner step is unnecessary in the univariate setting, as the complete graph Stein discrepancy $\mathcal{S}(Q, \mathcal{T}_P, \mathcal{G}_{\|\cdot\|_1, Q, G_1})$ can be computed directly by sorting the sample and boundary points and only enforcing constraints between consecutive points in this ordering. Thus, the complete graph Stein discrepancy is our recommended quality measure when $d = 1$, and a recipe for its computation is given in Algorithm 2.

---

**Algorithm 1** Multivariate Spanner Stein Discrepancy

---

**input:** $Q$, coordinate bounds $(\alpha_1, \beta_1), \ldots, (\alpha_d, \beta_d)$ with $-\infty \leq \alpha_j < \beta_j \leq \infty$ for all $j$
$G_2 \leftarrow$ Compute sparse 2-spanner of $\text{supp}(Q)$
**for** $j = 1$ **to** $d$ **do (in parallel)**
  $r_j \leftarrow$ Solve $j$-th coordinate linear program (7) with graph $G_2$ and boundary constraints (8)
**return** $\sum_{j=1}^{d} r_j$

---

**Algorithm 2** Univariate Complete Graph Stein Discrepancy

---

**input:** $Q$, bounds $(\alpha, \beta)$ with $-\infty \leq \alpha < \beta \leq \infty$
$(x_{(1)}, \ldots, x_{(n')}) \leftarrow \text{SORT}(\{x_1, \ldots, x_n, \alpha, \beta\} \cap \mathbb{R})$
**return** $\sup_{\gamma \in \mathbb{R}^{n'}, \Gamma \in \mathbb{R}^{n'}} \sum_{i=1}^{n'} q(x_{(i)})(\gamma_i \frac{d}{dx} \log p(x_{(i)}) + \Gamma_i)$

$\quad s.t.\ \|\Gamma\|_\infty \leq 1, \forall i \leq n', |\gamma_i| \leq \mathbb{I}\big[\alpha < x_{(i)} < \beta\big],\ \text{and},\ \forall i < n',$

$\quad \max\left( \frac{|\gamma_i - \gamma_{i+1}|}{x_{(i+1)} - x_{(i)}}, \frac{|\Gamma_i - \Gamma_{i+1}|}{x_{(i+1)} - x_{(i)}}, \frac{|\gamma_i - \gamma_{i+1} - \Gamma_i(x_{(i)} - x_{(i+1)})|}{\frac{1}{2}(x_{(i+1)} - x_{(i)})^2}, \frac{|\gamma_i - \gamma_{i+1} - \Gamma_{i+1}(x_{(i)} - x_{(i+1)})|}{\frac{1}{2}(x_{(i+1)} - x_{(i)})^2} \right) \leq 1$

---

## 5 Experiments

We now turn to an empirical evaluation of our proposed quality measures. We compute all spanners using the efficient C++ greedy spanner implementation of Bouts et al. [19] and solve all optimization programs using Julia for Mathematical Programming [20] with the default Gurobi 6.0.4 solver [21]. All reported timings are obtained using a single core of an Intel Xeon CPU E5-2650 v2 @ 2.60GHz.

### 5.1 A Simple Example

We begin with a simple example to illuminate a few properties of the Stein diagnostic. For the target $P = \mathcal{N}(0, 1)$, we generate a sequence of sample points i.i.d. from the target and a second sequence i.i.d. from a scaled Student's t distribution with matching variance and 10 degrees of freedom. The left panel of Figure 1 shows that the complete graph Stein discrepancy applied to the first $n$ Gaussian sample points decays to zero at an $n^{-0.52}$ rate, while the discrepancy applied to the scaled Student's t sample remains bounded away from zero. The middle panel displays optimal Stein functions $g$ recovered by the Stein program for different sample sizes. Each $g$ yields a test function $h \triangleq \mathcal{T}_P g$, featured in the right panel, that best discriminates the sample $Q$ from the target $P$. Notably, the Student's t test functions exhibit relatively large magnitude values in the tails of the support.

### 5.2 Comparing Discrepancies

We show in Theorem 14 in the appendix that, when $d = 1$, the classical Stein discrepancy is the optimum of a convex quadratically constrained quadratic program with a linear objective, $O(n)$ variables, and $O(n)$ constraints. This offers the opportunity to directly compare the behavior of the graph and classical Stein discrepancies. We will also compare to the Wasserstein distance $d_{\mathcal{W}_{\|\cdot\|}}$,

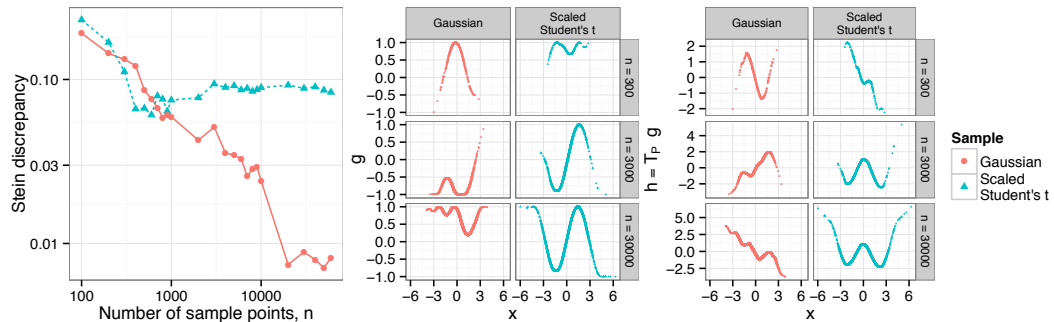

Figure 1: Left: Complete graph Stein discrepancy for a $\mathcal{N}(0, 1)$ target. Middle / right: Optimal Stein functions $g$ and discriminating test functions $h = \mathcal{T}_P g$ recovered by the Stein program.

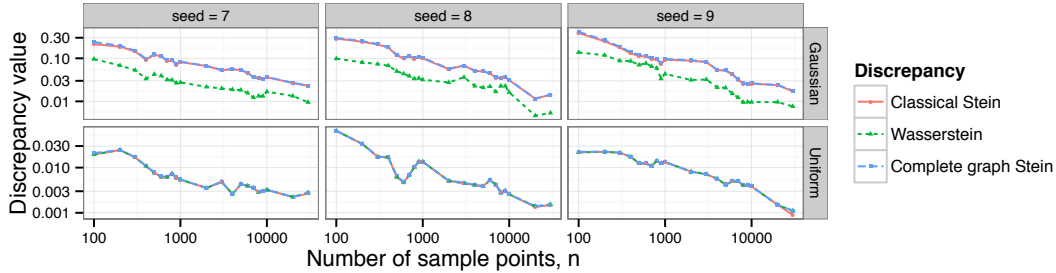

Figure 2: Comparison of discrepancy measures for sample sequences drawn i.i.d. from their targets.

which is computable for simple univariate target distributions [22] and provably lower bounds the non-uniform Stein discrepancies (5) with $c_{1:3} = (0.5, 0.5, 1)$ for $P = \text{Unif}(0, 1)$ and $c_{1:3} = (1, 4, 2)$ for $P = \mathcal{N}(0, 1)$ [9, 23]. For $\mathcal{N}(0, 1)$ and $\text{Unif}(0, 1)$ targets and several random number generator seeds, we generate a sequence of sample points i.i.d. from the target distribution and plot the non-uniform classical and complete graph Stein discrepancies and the Wasserstein distance as functions of the first $n$ sample points in Figure 2. Two apparent trends are that the graph Stein discrepancy very closely approximates the classical and that both Stein discrepancies track the fluctuations in Wasserstein distance even when a magnitude separation exists. In the $\text{Unif}(0, 1)$ case, the Wasserstein distance in fact equals the classical Stein discrepancy because $\mathcal{T}_P g = g'$ is a Lipschitz function.

## 5.3   Selecting Sampler Hyperparameters

Stochastic Gradient Langevin Dynamics (SGLD) [3] with constant step size $\epsilon$ is a biased MCMC procedure designed for scalable inference. It approximates the overdamped Langevin diffusion, but, because no Metropolis-Hastings (MH) correction is used, the stationary distribution of SGLD deviates increasingly from its target as $\epsilon$ grows. If $\epsilon$ is too small, however, SGLD explores the sample space too slowly. Hence, an appropriate choice of $\epsilon$ is critical for accurate posterior inference. To illustrate the value of the Stein diagnostic for this task, we adopt the bimodal Gaussian mixture model (GMM) posterior of [3] as our target. For a range of step sizes $\epsilon$, we use SGLD with minibatch size 5 to draw 50 independent sequences of length $n = 1000$, and we select the value of $\epsilon$ with the highest median quality – either the maximum effective sample size (ESS, a standard diagnostic based on autocorrelation [1]) or the minimum spanner Stein discrepancy – across these sequences. The average discrepancy computation consumes $0.4$s for spanner construction and $1.4$s per coordinate linear program. As seen in Figure 3a, ESS, which does not detect distributional bias, selects the largest step size presented to it, while the Stein discrepancy prefers an intermediate value. The rightmost plot of Figure 3b shows that a representative SGLD sample of size $n$ using the $\epsilon$ selected by ESS is greatly overdispersed; the leftmost is greatly underdispersed due to slow mixing. The middle sample, with $\epsilon$ selected by the Stein diagnostic, most closely resembles the true posterior.

## 5.4   Quantifying a Bias-Variance Trade-off

The approximate random walk MH (ARWMH) sampler [5] is a second biased MCMC procedure designed for scalable posterior inference. Its tolerance parameter $\epsilon$ controls the number of datapoint likelihood evaluations used to approximate the standard MH correction step. Qualitatively, a larger $\epsilon$ implies fewer likelihood computations, more rapid sampling, and a more rapid reduction of variance. A smaller $\epsilon$ yields a closer approximation to the MH correction and less bias in the sampler stationary distribution. We will use the Stein discrepancy to explicitly quantify this bias-variance trade-off.

We analyze a dataset of 53 prostate cancer patients with six binary predictors and a binary outcome indicating whether cancer has spread to surrounding lymph nodes [24]. Our target is the Bayesian logistic regression posterior [1] under a $\mathcal{N}(0, I)$ prior on the parameters. We run RWMH ($\epsilon = 0$) and ARWMH ($\epsilon = 0.1$ and batch size $= 2$) for $10^5$ likelihood evaluations, discard the points from the first $10^3$ evaluations, and thin the remaining points to sequences of length 1000. The discrepancy computation time for 1000 points averages $1.3$s for the spanner and $12$s for a coordinate LP. Figure 4 displays the spanner Stein discrepancy applied to the first $n$ points in each sequence as a function of the likelihood evaluation count. We see that the approximate sample is of higher Stein quality for smaller computational budgets but is eventually overtaken by the asymptotically exact sequence.

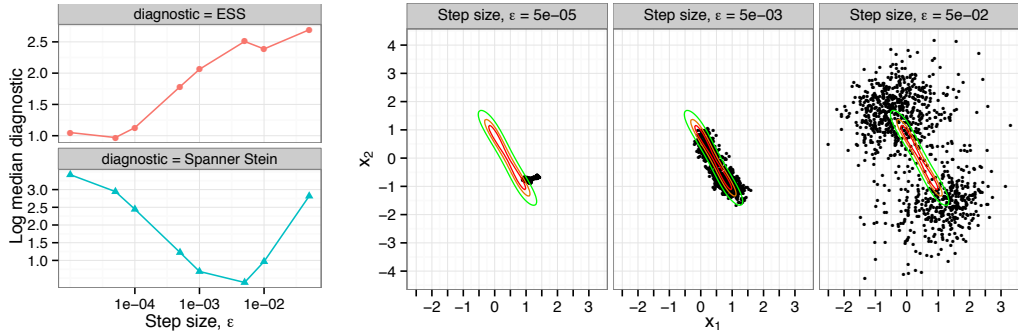

(a) Step size selection criteria      (b) 1000 SGLD sample points with equidensity contours of $p$ overlaid

Figure 3: (a) ESS maximized at $\epsilon = 5 \times 10^{-2}$; Stein discrepancy minimized at $\epsilon = 5 \times 10^{-3}$.

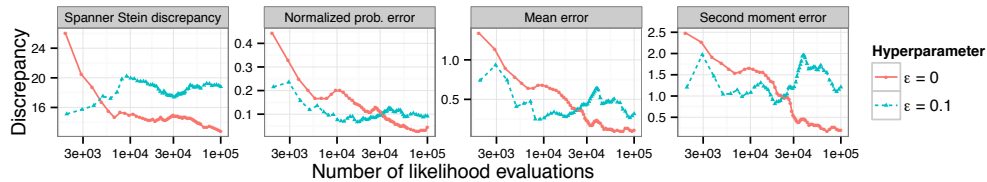

Figure 4: Bias-variance trade-off curves for Bayesian logistic regression with approximate RWMH.

To corroborate our result, we use a Metropolis-adjusted Langevin chain [25] of length $10^7$ as a surrogate $Q^*$ for the target and compute several error measures for each sample $Q$: normalized probability error $\max_l |\mathbb{E}[\sigma(\langle X, w_l \rangle) - \sigma(\langle Z, w_l \rangle)]| / \|w_l\|_\infty$, mean error $\frac{\max_j |\mathbb{E}[X_j - Z_j]|}{\max_j |\mathbb{E}_{Q^*}[Z_j]|}$, and second moment error $\frac{\max_{j,k} |\mathbb{E}[X_j X_k - Z_j Z_k]|}{\max_{j,k} |\mathbb{E}_{Q^*}[Z_j Z_k]|}$ for $X \sim Q$, $Z \sim Q^*$, $\sigma(t) \triangleq \frac{1}{1+e^{-t}}$, and $w_l$ the $l$-th datapoint covariate vector. The measures, also found in Figure 4, accord with the Stein discrepancy quantification.

## 5.5 Assessing Convergence Rates

The Stein discrepancy can also be used to assess the quality of deterministic sample sequences. In Figure 5 in the appendix, for $P = \text{Unif}(0,1)$, we plot the complete graph Stein discrepancies of the first $n$ points of an i.i.d. $\text{Unif}(0,1)$ sample, a deterministic Sobol sequence [26], and a deterministic kernel herding sequence [27] defined by the norm $\|h\|_{\mathcal{H}} = \int_0^1 (h'(x))^2 dx$. We use the median value over 50 sequences in the i.i.d. case and estimate the convergence rate for each sampler using the slope of the best least squares affine fit to each log-log plot. The discrepancy computation time averages 0.08s for $n = 200$ points, and the recovered rates of $n^{-0.49}$ and $n^{-1}$ for the i.i.d. and Sobol sequences accord with expected $O(1/\sqrt{n})$ and $O(\log(n)/n)$ bounds from the literature [28, 26]. As witnessed also in other metrics [29], the herding rate of $n^{-0.96}$ outpaces its best known bound of $d_{\mathcal{H}}(Q_n, P) = O(1/\sqrt{n})$, suggesting an opportunity for sharper analysis.

## 6 Discussion of Related Work

We have developed a quality measure suitable for comparing biased, exact, and deterministic sample sequences by exploiting an infinite class of known target functionals. The diagnostics of [30, 31] also account for asymptotic bias but lose discriminating power by considering only a finite collection of functionals. For example, for a $\mathcal{N}(0,1)$ target, the score statistic of [31] cannot distinguish two samples with equal first and second moments. Maximum mean discrepancy (MMD) on a characteristic Hilbert space [32] takes full distributional bias into account but is only viable when the expected kernel evaluations are easily computed under the target. One can approximate MMD, but this requires access to a separate trustworthy ground-truth sample from the target.

### Acknowledgments

The authors thank Madeleine Udell, Andreas Eberle, and Jessica Hwang for their pointers and feedback and Quirijn Bouts, Kevin Buchin, and Francis Bach for sharing their code and counsel.

## Footnotes

[1]One commonly considers real-valued functions $g$ when applying Stein's method, but we will find it more convenient to work with vector-valued $g$.

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
