[Supplementary Material]


[35] L. Mackey and J. Gorham. Multivariate Stein factors for strongly log-concave distributions. *arXiv:1512.07392*, December 2015.

Figure 5: Comparison of complete graph Stein discrepancy convergence for $P = \text{Unif}(0, 1)$.

# A   Proof of Proposition 1

Our integrability assumption together with the boundedness of $g$ and $\nabla g$ imply that $\mathbb{E}_P[\langle \nabla, g(Z) \rangle]$ and $\mathbb{E}_P[\langle g(Z), \nabla \log p(Z) \rangle]$ exist. Define the $\ell_\infty$ ball of radius $r$, $\mathbb{B}_r = \{x \in \mathbb{R}^d : \|x\|_\infty \le r\}$. Since $\mathcal{X}$ is convex, the intersection $\mathcal{X} \cap \mathbb{B}_r$ is compact and convex with Lipschitz boundary $\partial(\mathcal{X} \cap \mathbb{B}_r)$. Thus, the divergence theorem (integration by parts) implies that

$$\mathbb{E}_P[(\mathcal{T}_P g)(Z)] = \mathbb{E}_P[\langle \nabla, g(Z) \rangle + \langle g(Z), \nabla \log p(Z) \rangle] = \int_{\mathcal{X}} \langle \nabla, p(z) g(z) \rangle \, dz$$

$$= \lim_{r \to \infty} \int_{\mathcal{X} \cap \mathbb{B}_r} \langle \nabla, p(z) g(z) \rangle \, dz = \lim_{r \to \infty} \int_{\partial(\mathcal{X} \cap \mathbb{B}_r)} \langle g(z), n_r(z) \rangle p(z) \, dz$$

for $n_r$ the outward unit normal vector to $\partial(\mathcal{X} \cap \mathbb{B}_r)$. The final quantity in this expression equates to zero, as $\langle g(x), n(x) \rangle = 0$ for all $x$ on the boundary $\partial\mathcal{X}$, $g$ is bounded, and $\lim_{m \to \infty} p(x_m) = 0$ for any $(x_m)_{m=1}^\infty$ with $x_m \in \mathcal{X}$ for all $m$ and $\|x_m\|_\infty \to \infty$.

# B   Proof of Theorem 2: Stein Discrepancy Lower Bound for Strongly Log-concave Densities

We let $C^k(\mathcal{X})$ denote the set of real-valued functions on $\mathcal{X}$ with $k$ continuous derivatives and $d_{\mathcal{M}_{\|\cdot\|}}$ denote the *smooth function distance*, the IPM generated by

$$\mathcal{M}_{\|\cdot\|} \triangleq \left\{ h \in C^3(\mathcal{X}) \; \middle| \; \sup_{x \in \mathcal{X}} \max\left( \|\nabla h(x)\|^*, \|\nabla^2 h(x)\|^*, \|\nabla^3 h(x)\|^* \right) \le 1 \right\}.$$

We additionally define the operator norms $\|v\|_{op} \triangleq \|v\|_2$ for vectors $v \in \mathbb{R}^d$, $\|M\|_{op} \triangleq \sup_{v \in \mathbb{R}^d : \|v\|_2 = 1} \|Mv\|_2$ for matrices $M \in \mathbb{R}^{d \times d}$, and $\|T\|_{op} \triangleq \sup_{v \in \mathbb{R}^d : \|v\|_2 = 1} \|T[v]\|_{op}$ for tensors $T \in \mathbb{R}^{d \times d \times d}$.

The following result, proved in the companion paper [35], establishes the existence of explicit constants (*Stein factors*) $c_1, c_2, c_3 > 0$, such that, for any test function $h \in \mathcal{M}_{\|\cdot\|}$, the *Stein equation*

$$h(x) - \mathbb{E}_P[h(Z)] = (\mathcal{T}_P g_h)(x)$$

has a solution $g_h = \frac{1}{2} \nabla u_h$ belonging to the non-uniform Stein set $\mathcal{G}_{\|\cdot\|}^{c_{1:3}}$.

**Theorem 7** (Stein Factors for Strongly Log-concave Densities [35, Theorem 2.1])**.** *Suppose that $\mathcal{X} = \mathbb{R}^d$ and that $\log p \in C^4(\mathcal{X})$ is $k$-strongly concave with*

$$\sup_{z \in \mathcal{X}} \|\nabla^3 \log p(z)\|_{op} \le L_3 \quad and \quad \sup_{z \in \mathcal{X}} \|\nabla^4 \log p(z)\|_{op} \le L_4.$$

*For each $x \in \mathcal{X}$, let $(Z_{t,x})_{t \ge 0}$ represent the overdamped Langevin diffusion with infinitesimal generator*

$$(\mathcal{A}u)(x) = \frac{1}{2}\langle \nabla u(x), \nabla \log p(x) \rangle + \frac{1}{2}\langle \nabla, \nabla u(x) \rangle \tag{9}$$

*and initial state $Z_{0,x} = x$. Then, for each $h \in C^3(\mathcal{X})$ with bounded first, second, and third derivatives, the function*

$$u_h(x) \triangleq \int_0^\infty \mathbb{E}_P[h(Z)] - \mathbb{E}[h(Z_{t,x})] \, dt$$

*solves the the Stein equation*

$$h(x) - \mathbb{E}_P[h(Z)] = (\mathcal{A}u_h)(x) \tag{10}$$

*and satisfies*

$$\sup_{z \in \mathcal{X}} \|\nabla u_h(z)\|_2 \leq \frac{2}{k} \sup_{z \in \mathcal{X}} \|\nabla h(z)\|_2,$$

$$\sup_{z \in \mathcal{X}} \left\|\nabla^2 u_h(z)\right\|_{op} \leq \frac{2L_3}{k^2} \sup_{z \in \mathcal{X}} \|\nabla h(z)\|_2 + \frac{1}{k} \sup_{z \in \mathcal{X}} \left\|\nabla^2 h(z)\right\|_{op}, \text{ and}$$

$$\sup_{z,y \in \mathcal{X}, z \neq y} \frac{\left\|\nabla^2 u_h(z) - \nabla^2 u_h(y)\right\|_{op}}{\|z - y\|_2} \leq \frac{6L_3^2}{k^3} \sup_{z \in \mathcal{X}} \|\nabla h(z)\|_2 + \frac{L_4}{k^2} \sup_{z \in \mathcal{X}} \|\nabla h(z)\|_2$$

$$+ \frac{3L_3}{k^2} \sup_{z \in \mathcal{X}} \left\|\nabla^2 h(z)\right\|_{op} + \frac{2}{3k} \sup_{z \in \mathcal{X}} \left\|\nabla^3 h(z)\right\|_{op}.$$

Hence, by the equivalence of non-uniform Stein discrepancies (Proposition 4), $d_{\mathcal{M}_{\|\cdot\|}}(\mu, P) \leq \mathcal{S}(\mu, \mathcal{T}_P, \mathcal{G}_{\|\cdot\|}^{c_{1:3}}) \leq \max(c_1, c_2, c_3)\mathcal{S}(\mu, \mathcal{T}_P, \mathcal{G}_{\|\cdot\|})$ for any probability measure $\mu$.

The desired result now follows from Lemma 8, which implies that the Wasserstein distance $d_{\mathcal{W}_{\|\cdot\|}}(\mu_m, P) \to 0$ whenever $d_{\mathcal{M}_{\|\cdot\|}}(\mu_m, P) \to 0$ for a sequence of probability measures $(\mu_m)_{m \geq 1}$.

**Lemma 8** (Smooth-Wasserstein Inequality). *If $\mu$ and $\nu$ are probability measures on $\mathbb{R}^d$, and $\|v\| \geq \|v\|_2$ for all $v \in \mathbb{R}^d$, then*

$$d_{\mathcal{M}_{\|\cdot\|}}(\mu, \nu) \leq d_{\mathcal{W}_{\|\cdot\|}}(\mu, \nu) \leq 3 \max\left(d_{\mathcal{M}_{\|\cdot\|}}(\mu, \nu), \sqrt[3]{d_{\mathcal{M}_{\|\cdot\|}}(\mu, \nu)\sqrt{2}\,\mathbb{E}[\|G\|]^2}\right).$$

*for $G$ a standard normal random vector in $\mathbb{R}^d$.*

Lemma 2.2 of the companion paper [35] establishes this result for the case $\|\cdot\| = \|\cdot\|_2$; we omit the proof of the generalization which closely mirrors that of the Euclidean norm case.

## C  Proof of Proposition 3: Stein Discrepancy Upper Bound

Fix any $g$ in $\mathcal{G}_{\|\cdot\|}$. By Proposition 1, $\mathbb{E}[(\mathcal{T}_P g)(Z)] = 0$. The Lipschitz and boundedness contraints on $g$ and $\nabla g$ now yield

$$\begin{aligned}
\mathbb{E}_Q[(\mathcal{T}_P g)(X)] &= \mathbb{E}[(\mathcal{T}_P g)(X) - (\mathcal{T}_P g)(Z)] \\
&= \mathbb{E}[\langle g(X), \nabla \log p(X)\rangle - \langle g(Z), \nabla \log p(Z)\rangle + \langle \nabla, g(X) - g(Z)\rangle] \\
&= \mathbb{E}[\langle g(X), \nabla \log p(X) - \nabla \log p(Z)\rangle + \langle g(X) - g(Z), \nabla \log p(Z)\rangle] \\
&\quad + \mathbb{E}[\langle \nabla, g(X) - g(Z)\rangle] \\
&\leq \mathbb{E}[\|\nabla \log p(X) - \nabla \log p(Z)\|] + \mathbb{E}\big[\big\|\nabla \log p(Z)(X - Z)^\top\big\|\big] + \mathbb{E}[\|X - Z\|].
\end{aligned}$$

To derive the second advertised inequality, we use the definition of the matrix norm, the Fenchel-Young inequality for dual norms, the definition of the matrix dual norm, and the Cauchy-Schwarz inequality in turn:

$$\begin{aligned}
\mathbb{E}\big[\big\|\nabla \log p(Z)(X - Z)^\top\big\|\big] &= \mathbb{E}\left[\sup_{M:\|M\|^*=1} \langle \nabla \log p(Z), M(X - Z)\rangle\right] \\
&\leq \mathbb{E}\left[\sup_{M:\|M\|^*=1} \|\nabla \log p(Z)\|\|M(X - Z)\|^*\right] \\
&\leq \mathbb{E}[\|\nabla \log p(Z)\|\|X - Z\|] \leq \sqrt{\mathbb{E}\big[\|\nabla \log p(Z)\|^2\big]\mathbb{E}\big[\|X - Z\|^2\big]}.
\end{aligned}$$

Since our bounds hold uniformly for all $g$ in $\mathcal{G}_{\|\cdot\|}$, the proof is complete.

## D  Proof of Proposition 4: Equivalence of Non-uniform Stein Discrepancies

Fix any $c_1, c_2, c_3 > 0$, and let $c_{\max} = \max(c_1, c_2, c_3)$ and $c_{\min} = \min(c_1, c_2, c_3)$. Since the Stein discrepancy objective is linear in $g$, we have $a\,\mathcal{S}(Q, \mathcal{T}_P, \mathcal{G}_{\|\cdot\|}) = \mathcal{S}(Q, \mathcal{T}_P, a\mathcal{G}_{\|\cdot\|})$ for any $a > 0$. The result now follows from the observation that $c_{\min}\mathcal{G}_{\|\cdot\|} \subseteq \mathcal{G}_{\|\cdot\|}^{c_{1:3}} \subseteq c_{\max}\mathcal{G}_{\|\cdot\|}$.

## E  Proof of Proposition 5: Equivalence of Classical and Complete Graph Stein Discrepancies

The first inequality follows from the fact that $\mathcal{G}_{\|\cdot\|} \subseteq \mathcal{G}_{\|\cdot\|,Q,G_1}$. By the Whitney-Glaeser extension theorem [16, Thm. 1.4] of Glaeser [15], for every function $g \in \mathcal{G}_{\|\cdot\|,Q,G_1}$, there exists a function $\tilde{g} \in \kappa_d \mathcal{G}_{\|\cdot\|}^*$ with $g(x_i) = \tilde{g}(x_i)$ and $\nabla g(x_i) = \nabla\tilde{g}(x_i)$ for all $x_i$ in the support of $Q$. Here $\kappa_d$ is a constant, independent of $(Q, P)$, depending only on the dimension $d$ and norm $\|\cdot\|$. Since the Stein discrepancy objective is linear in $g$ and depends on $g$ only through the values $g(x_i)$ and $\nabla g(x_i)$, we have $\mathcal{S}(Q, \mathcal{T}_P, \mathcal{G}_{\|\cdot\|,Q,G_1}) \leq \mathcal{S}(Q, \mathcal{T}_P, \kappa_d\mathcal{G}_{\|\cdot\|}) = \kappa_d\,\mathcal{S}(Q, \mathcal{T}_P, \mathcal{G}_{\|\cdot\|})$.

## F  Proof of Proposition 6: Equivalence of Spanner and Complete Graph Stein Discrepancies

The first inequality follows from the fact that $\mathcal{G}_{\|\cdot\|,Q,G_1} \subseteq \mathcal{G}_{\|\cdot\|,Q,G_t}$. Fix any $g \in \mathcal{G}_{\|\cdot\|,Q,G_t}$ and any pair of points $z, z' \in \mathrm{supp}(Q)$. By the definition of $\mathcal{G}_{\|\cdot\|,Q,G_t}$, we have $\max\!\left(\|g(z)\|^*, \|\nabla g(z)\|^*\right) \leq 1$. By the $t$-spanner property, there exists a sequence of points $z_0, z_1, z_2, \ldots, z_{L-1}, z_L \in \mathrm{supp}(Q)$ with $z_0 = z$ and $z_L = z'$ for which $(z_{l-1}, z_l) \in E$ for all $1 \leq l \leq L$ and $\sum_{l=1}^{L}\|z_{l-1} - z_l\| \leq t\|z_0 - z_L\|$. Since $\max\!\left(\frac{\|g(z_{l-1}) - g(z_l)\|^*}{\|z_{l-1} - z_l\|}, \frac{\|\nabla g(z_{l-1}) - \nabla g(z_l)\|^*}{\|z_{l-1} - z_l\|}\right) \leq 1$ for each $l$, the triangle inequality implies that

$$\|\nabla g(z_0) - \nabla g(z_L)\|^* \leq \sum_{l=1}^{L}\|\nabla g(z_{l-1}) - \nabla g(z_l)\|^* \leq \sum_{l=1}^{L}\|z_{l-1} - z_l\| \leq t\|z_0 - z_L\|.$$

Identical reasoning establishes that $\|g(z_0) - g(z_L)\|^* \leq t\|z_0 - z_L\|$.

Furthermore, since $\|g(z_{l-1}) - g(z_l) - \nabla g(z_l)(z_{l-1} - z_l)\|^* \leq \frac{1}{2}\|z_{l-1} - z_l\|^2$ for each $l$, the triangle inequality and the definition of the tensor norm $\|\cdot\|^*$ imply that

$$\|g(z_0) - g(z_L) - \nabla g(z_L)(z_0 - z_L)\|^*$$

$$\leq \sum_{l=1}^{L}\|g(z_{l-1}) - g(z_l) - \nabla g(z_l)(z_{l-1} - z_l)\|^* + \|(\nabla g(z_l) - \nabla g(z_L))(z_{l-1} - z_l)\|^*$$

$$\leq \sum_{l=1}^{L}\frac{1}{2}\|z_{l-1} - z_l\|^2 + \|\nabla g(z_l) - \nabla g(z_L)\|^*\|z_{l-1} - z_l\|$$

$$\leq \sum_{l=1}^{L}\frac{1}{2}\|z_{l-1} - z_l\|^2 + \sum_{l'=l}^{L-1}\|\nabla g(z_{l'}) - \nabla g(z_{l'+1})\|^*\|z_{l-1} - z_l\|$$

$$\leq \sum_{l=1}^{L}\|z_{l-1} - z_l\|\left(\frac{1}{2}\|z_{l-1} - z_l\| + \sum_{l'=l}^{L-1}\|z_{l'} - z_{l'+1}\|\right) \leq \left(\sum_{l=1}^{L}\|z_{l-1} - z_l\|\right)^2 \leq t^2\|z_0 - z_L\|^2.$$

Since $z, z'$ were arbitrary, and the Stein discrepancy objective is linear in $g$, we conclude that $\mathcal{S}(Q, \mathcal{T}_P, \mathcal{G}_{\|\cdot\|,Q,G_t}) \leq \mathcal{S}(Q, \mathcal{T}_P, 2t^2\mathcal{G}_{\|\cdot\|,Q,G_1}) = 2t^2\,\mathcal{S}(Q, \mathcal{T}_P, \mathcal{G}_{\|\cdot\|,Q,G_1})$.

## G  Finite-dimensional Classical Stein Program

**Theorem 9** (Finite-dimensional Classical Stein Program). *If $\mathcal{X} = (\alpha, \beta)$ for $-\infty \leq \alpha < \beta \leq \infty$, and $x_{(1)} < \cdots < x_{(n')}$ represent the sorted values of $\{x_1, \ldots, x_n, \alpha, \beta\} \cap \mathbb{R}$, then the non-uniform*

*classical Stein discrepancy $\mathcal{S}(Q, \mathcal{T}_P, \mathcal{G}_{\|\cdot\|}^{c_{1:3}})$ is the optimal value of the convex program*

$$\max_{g} \quad \sum_{i=1}^{n'} q(x_{(i)}) \tfrac{d}{dx} \log p(x_{(i)}) g(x_{(i)}) + q(x_{(i)}) g'(x_{(i)}) \tag{11a}$$

$$\text{s.t.} \quad \forall i \in \{1, \dots, n'-1\}, \ |g'(x_{(i)})| \leq c_2, \ |g(x_{(i+1)}) - g(x_{(i)})| \leq c_2(x_{(i+1)} - x_{(i)}), \tag{11b}$$

$$g(x_{(i)}) - g(x_{(i+1)}) + \frac{1}{4c_3}\big(g'(x_{(i)}) - g'(x_{(i+1)})\big)^2 + \frac{x_{(i+1)} - x_{(i)}}{2}\big(g'(x_{(i)}) + g'(x_{(i+1)})\big)$$

$$+ \frac{1}{c_3}(L_b)_+^2 \leq \frac{c_3}{4}(x_{(i+1)} - x_{(i)})^2, \tag{11c}$$

$$g(x_{(i+1)}) - g(x_{(i)}) + \frac{1}{4c_3}\big(g'(x_{(i)}) - g'(x_{(i+1)})\big)^2 - \frac{x_{(i+1)} - x_{(i)}}{2}\big(g'(x_{(i)}) + g'(x_{(i+1)})\big)$$

$$+ \frac{1}{c_3}(L_u)_+^2 \leq \frac{c_3}{4}(x_{(i+1)} - x_{(i)})^2, \quad \text{and} \tag{11d}$$

$$\forall i \in \{1, \dots, n'\}, |g(x_{(i)})| \leq \mathbb{I}\big[\alpha < x_{(i)} < \beta\big]\big(c_1 - \frac{1}{2c_3}g'(x_{(i)})^2\big) \tag{11e}$$

*where $(r)_+ \triangleq \max(r, 0)$,*

$$L_b \triangleq \tfrac{c_3}{2}(x_{(i+1)} - x_{(i)}) - \tfrac{1}{2}\big(g'(x_{(i)}) + g'(x_{(i+1)})\big) - c_2, \quad \text{and}$$
$$L_u \triangleq \tfrac{c_3}{2}(x_{(i+1)} - x_{(i)}) + \tfrac{1}{2}\big(g'(x_{(i)}) + g'(x_{(i+1)})\big) - c_2.$$

We say the program (11) is finite-dimensional, because it suffices to optimize over vectors $\gamma, \Gamma \in \mathbb{R}^{n'}$ representing the function values ($\gamma_i = g(x_{(i)})$) and derivative values ($\Gamma_i = g'(x_{(i)})$) at each sample or boundary point $x_{(i)}$. Indeed, by introducing slack variables, this program is representable as a convex quadratically constrained quadratic program with $O(n)$ constraints, $O(n)$ variables, and a linear objective. Moreover, the pairwise constraints in this program are only enforced between neighboring points in the sequence of ordered locations $x_{(i)}$. Hence the resulting constraint matrix is sparse and banded, making the problem particularly amenable to efficient optimization.

**Proof** Throughout, we say that $\tilde{g}$ is an extension of $g$ if $\tilde{g}(x_{(i)}) = g(x_{(i)})$ and $\tilde{g}'(x_{(i)}) = g'(x_{(i)})$ for each $x_{(i)} \in \text{supp}(Q)$. Since the Stein objective only depends on $g$ and $g'$ through their values at sample points, $g$ and any extension $\tilde{g}$ have identical objective values.

We will establish our result by showing that every $g \in \mathcal{G}_{\|\cdot\|}^{c_{1:3}}$ is feasible for the program (11), so that $\mathcal{S}(Q, \mathcal{T}_P, \mathcal{G}_{\|\cdot\|}^{c_{1:3}})$ lower bounds the optimum of (11), and that every feasible $g$ for (11) has an extension in $\tilde{g} \in \mathcal{G}_{\|\cdot\|}^{c_{1:3}}$, so that $\mathcal{S}(Q, \mathcal{T}_P, \mathcal{G}_{\|\cdot\|}^{c_{1:3}})$ also upper bounds the optimum of (11).

### G.1 Feasibility of $\mathcal{G}_{\|\cdot\|}^{c_{1:3}}$

Fix any $g \in \mathcal{G}_{\|\cdot\|}^{c_{1:3}}$. Also, since $g'$ is $c_2$-bounded and $c_3$-Lipschitz, the constraints (11b) must be satisfied. Consider now the $c_2$-bounded and $c_3$-Lipschitz extensions of $g'$

$$B(t) \triangleq \max(-c_2, \max_{1 \leq i \leq n'} \big[g'(x_{(i)}) - c_3|t - x_{(i)}|\big]) \quad \text{and}$$

$$U(t) \triangleq \min(c_2, \min_{1 \leq i \leq n'} \big[g'(x_{(i)}) + c_3|t - x_{(i)}|\big]).$$

We know that $B(t) \leq g'(t) \leq U(t)$ for all $t$, for, if not, there would be a point $t_0$ and a point $x_{(i)}$ such that $|g'(x_{(i)}) - g'(t_0)| > c_3|x_{(i)} - t_0|$, which combined with the $c_3$-Lipschitz property would be a contradiction. Thus, for each sample $x_{(i)}$, the fundamental theorem of calculus gives

$$g(x_{(i+1)}) - g(x_{(i)}) = \int_{x_{(i)}}^{x_{(i+1)}} g'(t) \, dt \geq \int_{x_{(i)}}^{x_{(i+1)}} B(t) \, dt.$$

The right-hand side of this inequality evaluates precisely to the right-hand side of the constraint (11c). An analogous upper bound using $U(t)$ yields (11d).

Finally, consider any point $x_{(i)}$. If $x_{(i)} \in \{\alpha, \beta\}$, then (11e) is satisfied as $g(z) = 0$ for any point $z$ on the boundary. Suppose instead that $\alpha < x_{(i)} < \beta$. Without loss of generality, we may assume

that $g'(x_{(i)}) \geq 0$. Since $g'$ is $c_3$-Lipschitz, we have $g'(t) \geq g'(x_{(i)}) - c_3|t - x_{(i)}|$ for all $t$. Integrating both sides of this inequality from $x_{(i)}$ to $x_u = x_{(i)} + g'(x_{(i)})/c_3$, we obtain

$$g(x_u) - g(x_{(i)}) = \int_{x_{(i)}}^{x_u} g'(t)\, dt \geq \int_{x_{(i)}}^{x_u} g'(x_{(i)}) - c_3(t - x_{(i)})\, dt = g'(x_{(i)})^2/(2c_3)$$

Since $g(x_u) \leq c_1$, we have $\frac{1}{2c_3}g'(x_{(i)})^2 + g(x_{(i)}) \leq c_1$. Similarly, by integrating the inequality from $x_b = x_{(i)} - g'(x_{(i)})/c_3$ to $x_{(i)}$, we have $g(x_b) - g(x_{(i)}) \geq g'(x_{(i)})^2/(2c_3)$, which combined with $g(x_b) \leq c_1$ yields (11e).

## G.2 Extending Feasible Solutions

Suppose now that $g$ is any function feasible for the program (11). We will construct an extension $\tilde{g} \in \mathcal{G}_{\|\cdot\|}^{c_{1:3}}$ by first working independently over each interval $(x_{(i)}, x_{(i+1)})$. Fix an index $i < n'$. Our strategy is to identify a pair of $c_2$-bounded, $c_3$-Lipschitz functions $m_i$ and $M_i$ defined on the interval $[x_{(i)}, x_{(i+1)}]$ which satisfy $m_i(x) \leq M_i(x)$ for all $x \in [x_{(i)}, x_{(i+1)}]$, $m_i(x) = M_i(x) = g'(x)$ for $x \in \{x_{(i)}, x_{(i+1)}\}$, and $\int_{x_{(i)}}^{x_{(i+1)}} m_i(t)dt \leq g(x_{(i+1)}) - g(x_{(i)}) \leq \int_{x_{(i)}}^{x_{(i+1)}} M_i(t)dt$. For any such $(m_i, M_i)$ pair, there exists $\zeta_i \in [0, 1]$ satisfying

$$g(x_{(i+1)}) - g(x_{(i)}) = \int_{x_{(i)}}^{x_{(i+1)}} \zeta_i m_i(t) + (1 - \zeta_i)M_i(t)dt,$$

and hence we will define the extension

$$\tilde{g}(x) = g(x_{(i)}) + \int_{x_{(i)}}^{x} \zeta_i m_i(t) + (1 - \zeta_i)M_i(t)dt.$$

By convexity, the extension derivative $\tilde{g}'$ is $c_2$-bounded and $c_3$-Lipschitz, so we will only need to check that $\sup_{x \in \mathcal{X}} |\tilde{g}(x)| \leq c_1$. The maximum magnitude values of $\tilde{g}$ occur either at the interval endpoints, which are $c_1$-bounded by (11e), or at critical points $x$ satisfying $\tilde{g}'(x) = 0$, so it suffices to ensure that $\tilde{g}$ is $c_1$-bounded at all critical points.

We will use the $c_2$-bounded, $c_3$-Lipschitz functions $B$ and $U$ as building blocks for our extension, since they satisfy $B(t) = U(t) = g'(t)$ for $t \in \{x_{(i)}, x_{(i+1)}\}$ and $B(t) \leq g'(t) \leq U(t)$,

$$B(t) = \max(-c_2, g'(x_{(i)}) - c_3(t - x_{(i)}), g'(x_{(i+1)}) - c_3(x_{(i+1)} - t)), \quad \text{and}$$
$$U(t) = \min(c_2, g'(x_{(i)}) + c_3(t - x_{(i)}), g'(x_{(i+1)}) + c_3(x_{(i+1)} - t)),$$

for $t \in [x_{(i)}, x_{(i+1)}]$. We need only consider three cases.

**Case 1: $B$ and $U$ are never negative or never positive on $[x_{(i)}, x_{(i+1)}]$.** For this case, we will choose $m_i = B$ and $M_i = U$. By (11c) and (11d) we know $\int_{x_{(i)}}^{x_{(i+1)}} m_i(t)dt \leq g(x_{(i+1)}) - g(x_{(i)}) \leq \int_{x_{(i)}}^{x_{(i+1)}} M_i(t)dt$. Since $B$ and $U$ never change signs, $\tilde{g}$ will be monotonic and hence $c_1$-bounded for any choice of $\zeta_i$.

**Case 2: Exactly one of $B$ and $U$ changes sign on $[x_{(i)}, x_{(i+1)}]$.** Without loss of generality, we may assume that $g'(x_{(i)}), g'(x_{(i+1)}) \geq 0$ and that $B$ changes sign. Consider the quantity $\phi \triangleq \int_{x_{(i)}}^{x_{(i+1)}} \max\{B(t), 0\}dt$. If $g(x_{(i+1)}) - g(x_{(i)}) \leq \phi$, we let $m_i = B$ and $M_i = \max\{B, 0\}$.

Since, on the interval $[x_{(i)}, x_{(i+1)}]$, $B$ is piecewise linear with at most two pieces that can take on the value 0, $B$ has at most two roots within this interval. However, since $B(x)$ is continuous, negative for some value of $x$, and nonnegative at $x \in \{x_{(i)}, x_{(i+1)}\}$, we know $B$ has at least two roots. Thus let $r_1 < r_2$ be the roots of $B(x)$. For any choice of $\zeta_i$, the convex combination $\zeta_i m_i + (1 - \zeta_i)M_i$ will be exactly $B$ outside $(r_1, r_2)$. Moreover, if $\zeta_i \neq 0$, then this combination will be less than 0 on $(r_1, r_2)$, and if $\zeta_i = 0$, the combination will be 0 on the whole interval. Hence it suffices to only check the critical points $r_1$ and $r_2$. By (11e), $m_i(r) = M_i(r) = B(r) \in [-c_1, c_1]$ for $r \in \{r_1, r_2\}$, and so $\tilde{g}$ will be $c_1$-bounded.

If instead $g(x_{(i+1)}) - g(x_{(i)}) > \phi$, we can recycle the argument from Case 1 with $m_i = \max\{B, 0\}$ and $M_i = U$ and conclude that $\tilde{g}$ is $c_1$-bounded.

**Case 3: Both $B$ and $U$ change sign on $[x_{(i)}, x_{(i+1)}]$.** Without loss of generality, we may assume that $g'(x_{(i)}) \geq 0, g'(x_{(i+1)}) < 0$. Since $B$ continuously interpolates between $g'(x_{(i)})$ and $g'(x_{(i+1)})$ on $[x_{(i)}, x_{(i+1)}]$, it must have a root $r$. Let $w_i \in [x_{(i)}, x_{(i+1)}]$ be the point where $B$ changes from one linear portion to another. Then because $B$ is monotonic on each linear portion, the fact that $B(w_i) \leq B(x_{(i+1)}) < 0$ means that $B$ cannot have a root between $[w_i, x_{(i+1)}]$ and hence has at most one root on $[x_{(i)}, x_{(i+1)}]$. Hence $r$ is the unique root of $B$.

In a similar fashion, let us define $s$ as the root of $U$, and since $B(x) \leq U(x)$ for all $x$, we have $s \geq r$. Define

$$W(x) \triangleq \begin{cases} B(x) & x \in [x_{(i)}, r) \\ 0 & x \in [r, s] \\ U(x) & t \in (s, y], \end{cases}$$

and $\psi \triangleq \int_{x_{(i)}}^{x_{(i+1)}} W(t)dt$. As in Case 2, we will consider two subcases. If $g(x_{(i+1)}) - g(x_{(i)}) \leq \psi$, we will let $m_i = B$ and $M_i = W$. By (11e), $m_i(r) = M_i(r) = B(r) \in [-c_1, c_1]$, and since this is the only critical point, $\tilde{g}$ will be $c_1$-bounded.

For the other case, in which $g(x_{(i+1)}) - g(x_{(i)}) > \psi$, we choose $m_i = W$ and $M_i = U$. Then (11e) imply that $m_i(s) = M_i(s) = U(s) \in [-c_1, c_1]$, and, since this is the only critical point, the extension is well-defined on $(x_{(i)}, x_{(i+1)})$.

**Defining $\tilde{g}$ outside of the interval $[x_1, x_{n'}]$** It only remains to define our extension $\tilde{g}$ outside of the interval $[x_1, x_{n'}]$ when either $\alpha$ or $\beta$ is infinite. Suppose $\alpha = -\infty$. We extend $\tilde{g}$ to each $x \in (-\infty, x_1)$ using the construction

$$\tilde{g}(x) \triangleq \int_{-\infty}^{x} \mathbb{I}[t \in (x_1 - |g'(x_1)|/c_3, x_1)](g'(x_1) - c_3 \operatorname{sign}(g'(x_1))t) \, dt.$$

This extension ensures that $\tilde{g}'$ is $c_2$-bounded and $c_3$-Lipschitz. Moreover, the constraint (11e) guarantees that $|\tilde{g}(x)| \leq c_1$. Analogous reasoning establishes an extension to $(x_{n'}, \infty)$. $\qquad \square$

# H   Equivalence of Constrained Classical and Spanner Stein Discrepancies

For $P$ with support $\mathcal{X} = (\alpha_1, \beta_1) \times \cdots \times (\alpha_d, \beta_d)$ for $-\infty \leq \alpha_j < \beta_j \leq \infty$, Algorithm 1 computes a Stein discrepancy based on the graph Stein set

$$\mathcal{G}_{\|\cdot\|_1, Q, (V,E)} \triangleq \Big\{ g : \mathcal{X} \to \mathbb{R}^d \mid \forall x \in V, \ j, k \in \{1, \dots, d\} \text{ with } k \neq j, \text{ and } b_j \in \{\alpha_j, \beta_j\} \cap \mathbb{R},$$

$$\max\Big(\|g(x)\|_\infty, \|\nabla g(x)\|_\infty, \frac{|g_j(x)|}{|x_j - b_j|}, \frac{|\nabla_k g_j(x)|}{|x_j - b_j|}, \frac{|g_j(x) - \nabla_j g_j(x)(x_j - b_j)|}{\frac{1}{2}(x_j - b_j)^2}\Big) \leq 1, \text{ and, } \forall (x,y) \in E,$$

$$\max\Big(\frac{\|g(x)-g(y)\|_\infty}{\|x-y\|_1}, \frac{\|\nabla g(x)-\nabla g(y)\|_\infty}{\|x-y\|_1}, \frac{\|g(x)-g(y)-\nabla g(x)(x-y)\|_\infty}{\frac{1}{2}\|x-y\|_1^2}, \frac{\|g(x)-g(y)-\nabla g(y)(x-y)\|_\infty}{\frac{1}{2}\|x-y\|_1^2}\Big) \leq 1 \Big\},$$

Our next result shows that the graph Stein discrepancy based on a $t$-spanner is strongly equivalent to the classical Stein discrepancy.

**Proposition 10** (Equivalence of Constrained Classical and Spanner Stein Discrepancies)**.** *If $\mathcal{X} = (\alpha_1, \beta_1) \times \cdots \times (\alpha_d, \beta_d)$, and $G_t = (\operatorname{supp}(Q), E)$ is a $t$-spanner, then*

$$\mathcal{S}(Q, \mathcal{T}_P, \mathcal{G}_{\|\cdot\|_1}) \leq \mathcal{S}(Q, \mathcal{T}_P, \mathcal{G}_{\|\cdot\|_1, Q, G_t}) \leq t^2 \kappa_d \, \mathcal{S}(Q, \mathcal{T}_P, \mathcal{G}_{\|\cdot\|_1}),$$

*where $\kappa_d$ is a constant, independent of $(Q, P, G_t, t)$, depending only on the dimension $d$.*

**Proof**

**Establishing the first inequality** Fix any $g \in \mathcal{G}_{\|\cdot\|_1}$, $z \in \operatorname{supp}(Q)$, and $j, k \in \{1, \dots, d\}$ with $k \neq j$, and consider any $j$-th coordinate boundary projection point

$$b \in \{z + e_j(\alpha_j - z_j), z + e_j(\beta_j - z_j)\} \cap \mathbb{R}^d.$$

Since $b \in \partial \mathcal{X}$, we must have $\langle g(b), n(b) \rangle = \langle g(b), e_j \rangle = g_j(b) = 0$. Moreover, for each dimension $k \neq j$, we have $\nabla_k g_j(x) = 0$, since otherwise, $\langle g(b + \delta e_k), n(b + \delta e_k) \rangle = g_j(b + \delta e_k) \neq 0$ for some $\delta \in \mathbb{R}$ and $b + \delta e_k \in \partial \mathcal{X}$ by the continuity of $\nabla g_j$.

The smoothness constraints of the classical Stein set $\mathcal{G}_{\|\cdot\|_1}$ now imply that

$$|g_j(z)| = |g_j(z) - g_j(b)| \leq |z_j - b_j|, \quad |\nabla_k g_j(x)| = |\nabla_k g_j(z) - \nabla_k g_j(b)| \leq |z_j - b_j|,$$

and

$$|g_j(z) - \nabla_j g_j(x)(z_j - b_j)| = |g_j(b) - g_j(z) - \langle \nabla g_j(z), b - z \rangle| \leq \frac{1}{2}(z_j - b_j)^2$$

so that all graph Stein set boundary compatibility constraints are satisfied. Hence, we have the containment $\mathcal{G}_{\|\cdot\|_1} \subseteq \mathcal{G}_{\|\cdot\|_1, Q, G_t}$, which implies the first advertised inequality.

**Establishing the second inequality**   To establish the second inequality, it suffices to show that for any $\tilde{g} \in \mathcal{G}_{\|\cdot\|_1, Q, G_t}$, each $j \in \{1, \ldots, d\}$, and $\zeta \triangleq t$, there exists a function $g_j$ satisfying

$$g_j(z) = \tilde{g}_j(z), \ \nabla g_j(z) = \nabla \tilde{g}_j(z), \ g_j(b) = 0, \ \nabla_k g_j(b) = 0, \ \forall k \neq j, \tag{12}$$
$$|g_j(b) - g_j(z)| \leq \|b - z\|_1, \tag{13}$$
$$\|\nabla g_j(b) - \nabla g_j(z)\|_\infty \leq \zeta \|b - z\|_1, \ \|\nabla g_j(b) - \nabla g_j(b')\|_\infty \leq \zeta \|b - b'\|_1, \tag{14}$$
$$|g_j(b) - g_j(z) - \langle \nabla g_j(z), b - z \rangle| \leq \frac{\zeta}{2} \|b - z\|_1^2, \tag{15}$$
$$|g_j(z) - g_j(b) - \langle \nabla g_j(b), z - b \rangle| \leq \frac{3\zeta}{2} \|b - z\|_1^2, \quad \text{and} \tag{16}$$
$$|g_j(b) - g_j(b') - \langle \nabla g_j(b'), b - b' \rangle| \leq \frac{\zeta}{2} \|b - b'\|_1^2 \tag{17}$$

for all $z \in \mathrm{supp}(Q)$ and all $b, b'$ in the $j$-th coordinate boundary set

$$B_j \triangleq \{b \in \mathbb{R}^d : b = z + e_j(\alpha_j - z_j) \text{ or } b = z + e_j(\beta_j - z_j) \text{ for some } z \in \mathcal{X}\}.$$

Indeed, since such $g_j$ will satisfy $\max(|g_j(z)|, \|\nabla g_j(z)\|_\infty) \leq 1$ for all $z \in \mathrm{supp}(Q) \cup B_j$ and

$$\max\left( \frac{|g_j(x) - g_j(y)|}{\|x - y\|_1}, \frac{\|\nabla g_j(x) - \nabla g_j(y)\|_\infty}{\|x - y\|_1}, \frac{|g_j(x) - g_j(y) - \nabla g_j(x)(x - y)|}{\frac{1}{2}\|x - y\|_1^2}, \frac{|g_j(x) - g_j(y) - \nabla g_j(y)(x - y)|}{\frac{1}{2}\|x - y\|_1^2} \right) \leq 2t^2$$

for all $x, y \in \mathrm{supp}(Q)$ by the argument of Appendix F, the Whitney-Glaeser extension theorem [16, Thm. 1.4] of Glaeser [15] will then imply that there exists $g^* \in t^2 \kappa_d \, \mathcal{G}_{\|\cdot\|_1}$, for a constant $\kappa_d$ independent of $\tilde{g}$ depending only on $d$, with $g^*(z) = g(z)$ and $\nabla g^*(z) = \nabla g(z)$ for all $z \in \mathrm{supp}(Q)$. Since $\tilde{g}$ and $g^*$ will have matching Stein discrepancy objective values, and each objective is linear in $g$, the second advertised inequality will then follow.

Fix $\tilde{g} \in \mathcal{G}_{\|\cdot\|_1, Q, G_t}$ and $j \in \{1, \ldots, d\}$. We will now construct a function $g_j$ satisfying the desired properties. Since $g_j$ and $\nabla g_j$ are determined on $\mathrm{supp}(Q)$, and $g_j$ and $\nabla_k g_j$ are determined on $B_j$ for $k \neq j$ by the constraints (12), it remains to define $\nabla_j g_j$ on $B_j$. We choose the extension

$$\nabla_j g_j(b) \triangleq \min_{z \in \mathrm{supp}(Q)} \{\nabla_j g_j(z) + \zeta \|z - b\|_1\} \quad \text{for all} \quad b \in B_j.$$

Fix any $z \in \mathrm{supp}(Q)$ and $b \in B_j$, and let $b^* = z + e_j(b_j - z_j)$. The argument of Appendix F implies that $\nabla_j g_j$ is $\zeta$-Lipschitz on $\mathrm{supp}(Q)$, and hence it is also $\zeta$-Lipschitz on $\mathrm{supp}(Q) \cup B_j$. Since

$$|\nabla_k g_j(z) - \nabla_k g_j(b)| = |\nabla_k g_j(z)| \leq |z_j - b_j| \leq \|z - b\|_1$$

for all $k \neq j$, we have (14). Moreover, the boundary compatibility constraints of $\mathcal{G}_{\|\cdot\|_1, Q, G_t}$ imply

$$|g_j(b) - g_j(z)| = |g_j(z)| \leq \|b^* - z\|_1 \leq \|b - z\|_1,$$

establishing (13). We next invoke the triangle inequality, the boundary compatibility conditions of $\mathcal{G}_{\|\cdot\|_1, Q, G_t}$, Hölder's inequality, the Lipschitz derivative property (14), and the fact $\|z - b\|_1 =$

$\|b^* - z\|_1 + \|b^* - b\|_1$ in turn to establish (15):

$$
\begin{aligned}
|g_j(b) - g_j(z) - \langle \nabla g_j(z), b - z \rangle| &= |g_j(z) - \nabla_j g_j(z)(z_j - b_j) - \langle \nabla g_j(z), b^* - b \rangle| \\
&\leq |g_j(z) - \nabla_j g_j(z)(z_j - b_j)| + |\langle \nabla g_j(b^*) - \nabla g_j(z), b^* - b \rangle| \\
&\leq \frac{1}{2} \|b^* - z\|_1^2 + \|\nabla g_j(b^*) - \nabla g_j(z)\|_\infty \|b^* - b\|_1 \\
&\leq \frac{1}{2} \|b^* - z\|_1^2 + \zeta \|b^* - z\|_1 \|b^* - b\|_1 \\
&\leq \frac{\zeta}{2} (\|b^* - z\|_1 + \|b^* - b\|_1)^2 = \frac{\zeta}{2} \|b - z\|_1^2.
\end{aligned}
$$

A parallel argument yields (17). Finally, we may deduce (16), as

$$
\begin{aligned}
|g_j(z) - g_j(b) - \langle \nabla g_j(b), z - b \rangle| &\leq |g_j(z) - \nabla_j g_j(z)(z_j - b_j)| + |\nabla_j g_j(b) - \nabla_j g_j(z)||z_j - b_j| \\
&\leq \frac{1}{2} (z_j - b_j)^2 + \zeta \|b - z\|_1 |z_j - b_j| \leq \frac{3\zeta}{2} \|b - z\|_1^2
\end{aligned}
$$

by the triangle inequality, the definition of $\mathcal{G}_{\|\cdot\|_1, Q, G_t}$, and the Lipschitz property (14). $\qquad\square$