[Reviews · NeurIPS 2015]

Submitted by Assigned_Reviewer_1

This paper considers the general problem of estimating expectations $E_P h$ under an arbitrary sample drawn (effectively) from some $Q$, and in particular of a method for calculating the quality of $Q$ as a surrogate for $P$. In particular, they focus on integral probability metrics to measure the quality of $P$.

The paper includes interesting extensions of technical results on the Stein discrepancy, though at times it is unclear which ones are the authors' own contributions. But overall, the theoretical results seem thorough and complete.

The paper also includes a method for efficiently computing Stein discrepancies, which also seems novel, and whch they show is the same order as the classical Stein discrepancy.

Overall, the method is highly interesting and seems to have general applicability. The experimental results illustrate it well.

Summary: The paper presents theory and algorithms for estimating Stein discrepancies between distributions. The theoretical results are interesting and the method is highly applicable.

Submitted by Assigned_Reviewer_2

The main contribution of the paper is a proposal for IPM with certain properties and geared towards a specific application. The IPM proposed is based on Stein operator, which is one of the standard methods, by now, for studying convergence in distribution. The motivating application is measuring the quality of biased MCMC procedures. The assumption are: (i) the expectation under the target distribution is zero (which is made because of the Stein's method) (ii) the operator used is the infinitesimal generator of Markov process and (iii) the set of functions in defining the IPM is what is called as 'classic Stein set' in page 3 of the paper.

The main contributions of the paper are 1) Theorem 7 (and theorem 2) relating Stein and (Wasser)stein IPMs for log-concave densities under certain assumptions in the multivariate setting. 2) Providing computable optimization programs for Stein discrepancy - I especially liked the use of

Whitney's extension theorem for computing a graph based discrepancy.

It would be interesting to explore rates of convergence results for Stein discrepancy rigorously. Furthermore, it would be interesting to explore relationship to MMD (as one of the main focus in on computability). Using MMD in this context would correspond to goodness-of-fit style testing for target distribution (say P) using MMD. The authors mention MMD would work when 'expected kernel evaluations are easily computed under the target'. But since the space of rkhs function is dense in L_2(P) essentially it could approximate a large class of functions (not in the sample sense as mentioned in the paper but in a population setting).

I went over the proofs and they seem to be correct at least at a high level. The contributions of this paper are interesting and I would recommend accepting the paper. I think extending and exploring ideas along the lines of this paper should lead to interesting results.
Summary: The contributions of this paper are interesting and I would recommend accepting the paper. I think extending and exploring ideas along the lines of this paper should lead to interesting results.

Submitted by Assigned_Reviewer_3

Massive data pose many challenges to MC methods.

In a variety of applications it is easier to sample using an approximate transition kernel than the exact. Some examples include posterior exploration or maximum a posteriori estimation based on small mini-batches of the data. This paper introduces a pseudo-metric based on Stein's method to compare samples from the approximate and exact procedure.

Theoretically grounded, the method is applicable to wide variety of important problems and can be used efficiently using off-the-shelf linear programming solvers.

The paper is clear and well-written.

It introduces a new application of Stein's method that has excellent applications to current MC methods.

We also recommend a few suggestions for improvements:

- In the decoupled linear program why is \ell_1 norm appropriate? It would be beneficial to compare the simulation results for say \ell_1 and \ell_2 norms. - provide a few intuition about the sufficient conditions in Thm 2 and why is it not necessary; similar comment for definition of set G in Eqn (5).

Summary: This work introduces a pseudo-metric based on Stein's method to compare the Monte Carlo (MC) estimates from approximate methods and their target value.

The proposed approach is theoretically grounded and can be efficiently computed.

Many experiments are presented demonstrate the theoretical results on real and simulated data.

Submitted by Assigned_Reviewer_4

Although much more technically involved than your average NIPS paper, it was a pleasure reading and reviewing this manuscript. I do not have many comments:

-Section 4 compresses the material far too much, and makes it difficult to follow. It may be worthwhile to remove some of the results (or move them to an appendix), and expand the discussion in the text. For a journal extension I would certainly include all the detail presented here, but as this is a conference paper I would favour simplicity/clarity over detail.

-Some extra discussion of the implications of the various Stein functions in figure 1 would be nice. I am not sure what to extract from these given the text.

-I think the only thing that is sorely lacking in this paper is an evaluation (experimental, theoretical, or optimally both) of the computational complexity/practicality of what you've proposed. Please comment on why this is missing or suggest a change to the paper that addresses this.
Summary: This paper is a very cool application of Stein's method to assessing the quality of samples from MCMC algorithms. It addresses a major gap in state of the art MCMC research. While this paper should probably just be a journal publication (it certainly has enough material to warrant it), I would love to see this at NIPS.

Author Feedback
Author rebuttal: We thank the reviewers for their helpful feedback and provide our responses to specific points raised below.

Reviewer 1:

"Rates of convergence": A theoretical assessment of convergence rates would be very interesting. Prop. 3 (or other upper-bounding techniques) could be used to analyze Stein discrepancy convergence under various sampling assumptions. Thm. 7 and Lem. 8 would then imply explicit rates of convergence for the smooth function distance and Wasserstein distance, respectively.

"Relationship to MMD": MMD (in its most common use as an IPM with H the unit ball of a reproducing kernel Hilbert space) has a number of desirable properties: its IPM optimization problem has a closed-form solution, and, when a universal kernel like a Gaussian kernel is used, MMD satisfies Desiderata (i) and (ii). The chief drawback is that the MMD solution requires explicit integration under P. For example, Gaussian kernel MMD requires computing integrals of the form \int \exp(-||x-z||^2) p(z). While this is straightforward for select target distributions (like finite Gaussian mixtures), these integrals are unavailable for most targets of interest. This limitation on the applicability of MMD was one of the initial motivations for this work, and we will endeavor to make this important point clearer in the text. Note that one can avoid the intractable integration under P by approximating P with a sample from P, but this requires access to a separate surrogate sample.

Reviewer 2:

"Why the L1 norm?": The graph Stein discrepancy equipped with the L1 norm has two especially desirable properties: 1) the resulting optimization problem is a linear program and 2) the optimization problem decomposes into d independent linear programs that may be solved in parallel. Other norm choices, including L2, require solving a much larger coupled non-linear program instead of d decoupled smaller linear programs. We will work to make these points clearer in the text.

"Sufficient conditions in Thm. 2": The Thm. 2 conditions relate to the speed at which the overdamped Langevin diffusion underlying our Stein operator T_P converges to stationarity and were chosen to ensure that the diffusion was a sufficiently smooth function of its starting point. Recent work by Eberle ("Reflection coupling and Wasserstein contractivity without convexity
") suggests that these conditions can be relaxed, and alternative sufficient conditions (for the univariate case) appear in [9,10].

"Intuition about Eq. (5)": To establish Thm. 2, we first bound the Wasserstein distance by a Stein discrepancy based on the non-uniform Stein set in Eq. (5) and then bound the non-uniform Stein discrepancy by the classical Stein discrepancy (Prop. 4). It is in this sense that the non-uniform Stein set leads to a tighter bound on the Wasserstein distance. For general use with a new target distribution, we advocate the parameter-free classical Stein and graph Stein sets, so we do not use the non-uniform Stein sets in our applications.

Reviewer 3:

"Sec. 4 overly compressed ... may be worthwhile to remove some of the results": We are reluctant to cut an example, as we hope that the variety of examples will inspire members of the community with distinct interests to build upon these ideas. However, we agree that Sec. 4 is overly compressed, and we will prioritize revising this section to improve clarity.

"Implications of Stein functions": The recovered test function h = T_Pg represents an explicit witness of the discrepancy between Q and P (as the expectations of Q and P over h deviate by the Stein discrepancy). We are not yet comfortable drawing additional conclusions from these functions, but we believe that a thorough cataloguing of Stein functions for various sample-target pairs is warranted to better understand the implications.

"Evaluation of computational complexity/practicality of what you've proposed":
We will make space to report experiment timings for each of our proposed applications.
Using a single core of a dual socket Intel(R) Xeon(R) CPU E5-2650 v2 @ 2.60GHz, the most expensive Stein discrepancy computations had the following running times:
-Hyperparameter selection: 0.4s (spanner), 1.4s (coordinate LP)
-Bias-variance tradeoff: 1.3s (spanner), 11s (coordinate LP)
-Convergence rates: 0.08s (LP)